# Role of Extracellular Matrix in Pathophysiology of Patent Ductus Arteriosus: Emphasis on Vascular Remodeling

**DOI:** 10.3390/ijms21134761

**Published:** 2020-07-04

**Authors:** Ting-Yi Lin, Jwu-Lai Yeh, Jong-Hau Hsu

**Affiliations:** 1School of Medicine, Kaohsiung Medical University, Kaohsiung 80708, Taiwan; lintingyi2014@gmail.com; 2Graduate Institute of Medicine, College of Medicine, Kaohsiung Medical University, Kaohsiung 80708, Taiwan; jwulai@kmu.edu.tw; 3Department of Pharmacology, College of Medicine, Kaohsiung Medical University, Kaohsiung 80708, Taiwan; 4Department of Medical Research, Kaohsiung Medical University, Kaohsiung 80708, Taiwan; 5Department of Marine Biotechnology and Resources, National Sun Yat-sen University, Kaohsiung 80424, Taiwan; 6Department of Pediatrics, Kaohsiung Medical University Hospital, Kaohsiung Medical University, Kaohsiung 80708, Taiwan; 7Department of Pediatrics, Faculty of Medicine, College of Medicine, Kaohsiung Medical University, Kaohsiung 80708, Taiwan

**Keywords:** patent ductus arteriosus, remodeling, extracellular matrix, intimal thickening

## Abstract

The ductus arteriosus (DA) is a shunt vessel between the aorta and the pulmonary artery during the fetal period that is essential for the normal development of the fetus. Complete closure usually occurs after birth but the vessel might remain open in certain infants, as patent ductus arteriosus (PDA), causing morbidity or mortality. The mechanism of DA closure is a complex process involving an orchestration of cell–matrix interaction between smooth muscle cells (SMC), endothelial cells, and extracellular matrix (ECM). ECM is defined as the noncellular component secreted by cells that consists of macromolecules such as elastin, collagens, proteoglycan, hyaluronan, and noncollagenous glycoproteins. In addition to its role as a physical scaffold, ECM mediates diverse signaling that is critical in development, maintenance, and repair in the cardiovascular system. In this review, we aim to outline the current understandings of ECM and its role in the pathophysiology of PDA, with emphasis on DA remodeling and highlight future outlooks. The molecular diversity and plasticity of ECM present a rich array of potential therapeutic targets for the management of PDA.

## 1. Introduction

The ductus arteriosus (DA) is a shunt vessel between the aorta (Ao) and the pulmonary artery (PA) during the fetal period that is essential for the normal development of the fetus. The DA sometimes persists after birth and causes common clinical morbidity, especially in low-birthweight infants [1]. The blood from the high-pressure Ao shunts to the low-pressure PA (left to right shunt) and causes pulmonary edema and decreases systemic perfusion, notably renal, mesenteric, and cerebral circulation [2]. The increased hemodynamic burden brought by pulmonary overcirculation eventually results in congestive cardiac failure, as shown in Figure 1.

Current pharmacologic management mostly relies on the inhibition of prostaglandin (PG) synthesis, such as with indomethacin or ibuprofen [3]. However, this is not responsive in 25% of patients [4]. Moreover, despite advances in the understanding of patent ductus arteriosus (PDA) molecular pathogenesis, pathways mediated by extracellular matrix (ECM) for the regulation of DA closure are not fully understood. Further knowledge of PDA pathogenesis is necessary to dissect the complex cell–matrix crosstalk regulating DA closure [5].

## 2. DA: Mechanism of Closure

Patency of fetal DA is maintained by the vasodilatory effect of low fetal oxygen tension, and placental cyclooxygenase-mediated products [6,7]. Successful closure of the DA requires the reversal of these patency drivers during the transition from fetal to the neonatal period. Indeed, the closure mechanism is then effected in two phases: smooth muscle constriction (functional closure) within 18–24 h after birth, and remodeling of the intima (anatomical closure) over the next few days or weeks. In this review, we focus on the intimal remodeling that highlights the critical role that ECM plays to allow successful DA closure.

### 2.1. Functional Closure

Within 24–48 h of birth, the decrease of PGE2 is mediated by the now-functioning lung metabolizing PGE and the elimination of the placental source. The withdrawal of the PGE-induced vasodilation results in the contraction of the medial layer in the DA that results in lumen obliteration and ductal shortening. Consequently, the loss of luminal blood flow causes a zone of hypoxia in the muscle media that is responsible for the ultimate anatomical closure [8]. Moreover, the postdelivery induced abrupt increase in oxygen tension inhibits DA smooth muscle cell (DASMC) voltage-dependent potassium channels that generate an influx of calcium that mediates DASMC constriction [9].

### 2.2. Anatomical Closure

Within the next two to three weeks, morphological and molecular remodeling yield the obliteration of DA lumen. The hypoxic zone induces local SMC death in the media and the production of growth factors that stimulate neointimal thickening, fibrosis, and permanent closure. Furthermore, vessel wall hypoxia inhibits endogenous PGE and nitric oxide production and averts subsequent reopening [10]. The gross histological composition of fetal DA resembles that of the contiguous main PA and descending Ao. Distinctions lie within the media of the arteries. Whereas circumferentially arranged layers of elastic fibers are present in the large arteries, longitudinally and spirally arranged layers of smooth muscle fibers are present within loose, concentric layers of elastic tissues in DA. Additionally, the intima of the DA is thickened and irregular, with abundant mucoid that is referred to as intimal cushions. The intrinsic difference in ECM composition and the structuring of DA compared to that of Ao emphasize the critical role of ECM in the pathophysiology of PDA.

## 3. Role of ECM in the Cardiovascular System

Exploring the role of ECM in systemic vascular intimal thickening allows us to unravel the complexity of ECM function before diving into its role in PDA and translate recent discoveries to this disease model. ECM is defined as the noncellular component secreted by cells that consists of macromolecules such as elastin, collagens, proteoglycan, hyaluronan (HA), and noncollagenous glycoproteins (GP). Despite its function as a physical scaffold, ECM plays a role in diverse signaling that is critical in development, maintenance, and repair. Cell–matrix interactions not only encompass cell receptor binding properties such as adhesion, migration, proliferation, differentiation, and survival [11], but also accommodation of multiple proteins with growth factors that establishes chemotaxis gradient. The dynamic architecture bioactively interacts with cells and generates signals that allow adaptive responses of intracellular and extracellular compartments to control cellular behavior, phenotype, and milieu homeostasis. Constant remodeling between matrix deposition and matrix degradation by proteases and their intricate control of activation and inhibition afford a delicate balance that may be disturbed during vascular pathologies. Excessive accumulation of dysregulated ECM may sabotage its physiological function as well.

Furthermore, matrix remodeling may remain futile without successful assembly into the three-dimensional physiological organization. In Table 1, we outline the diverse function of ECM in systemic vasculature. In Table 2, we summarize the mediators that interact with ECM in systemic vascular intimal thickening that is not confined to DA. Similarities in intimal thickening may offer additional insights referenced from vascular pathologies in atherosclerosis and arterial injury.

## 4. Role of ECM in DA Remodeling

Remodeling of DA is essential to permanent anatomical closure. The process is complex, with several mechanisms including intimal cushion formation, SMC migration and proliferation, endothelial cell proliferation, blood cell interaction [30], and ECM production. In this review, we will discuss the fundamental role of ECM in the pathogenesis of PDA. In Table 3 and Table 4, we summarize the different roles of matrix elements in DA remodeling.

Diverse signaling pathways orchestrate the subsequent luminal DA remodeling where complete closure is sequentially mediated in four phases: the deposition of ECM in the subendothelial region, the disassembly of the internal elastic lamina (IEL), loss of elastic fiber in the medial layer, and followed by the migration of SMC into the subendothelial space for the formation of intima thickening. We summarize the key sequential steps of anatomical closure in Figure 2.

### 4.1. ECM Deposition in the Subendothelial Region

DA closure begins with ECM deposition in the subendothelial region (SR). SR is defined as the layer between the IEL and endothelial cells (EC). Initially, SR is composed of granular and amorphous materials without collagen fibrils and elastin. In the PDA, the ECs remain attached to the IEL. The composition of ECM is critical in determining the success of endothelial detachment that permits SR thickening. De Reeder et al. immunohistochemically studied the topography of the ECM components that act in the adherence of the EC to the underlying intimal layers: collagen type I, III, IV, fibronectin (FN), and laminin (LN).

Interestingly, the ECM profile alters significantly before and after the detachment of the ECs. Where LN and collagen type I are diffusely present before but absent after separation of the EC. Collagen type III, barely detectable prior to detachment, become visible underneath the detached cells [43]. Consistent with ECM’s role in mediating ductal closure, these observed alterations of the ECM profile were confined to DA regions that developed intimal thickening only. The finding fails to explain the underlying mechanism for the determination of EC detachment that has been previously reported in a study to be related to an increase in HA [43].

#### 4.1.1. Hyaluronan (HA)

A product of the PGE-EP4 axis, HA deposition, is critical in promoting SMC migration into the SR. Examining the ECM of the SR, De Reeder et al. found the thoroughly deposited HA in the entire region of closing DA was absent in PDA [36]. They suggested that the hygroscopic properties of HA might cause an influx of water that loosens and expands the SR, which promotes SMC migration [36]. It is well established that PGE_2_ plays a primary role in maintaining the patency of the DA via its receptor EP4; however, genetic disruption of the PGE pathways, such as genetically modified mice with receptor EP4 deficiency [44,45], and COX-1 and COX-2 disruption [46] both paradoxically result in fatal patent DA in mice. Diving into these findings, Yokoyama et al. have confirmed that patency of DA observed in EP4-disrupted neonatal mice results from the absence of intimal thickening [38]. Histologically, HA production was found to be markedly reduced in EP4-disrupted DA compared to that of the wild-type DA. They proposed that the PGE-EP4 induced HA secretion is mediated through the PGE_2_-EP4-cyclic AMP (cAMP)-protein kinase A (PKA) signaling axis that upregulates HA synthase type 2 mRNA [7,47]. Dissecting into the adenylyl cyclases (AC) responsible for HA secretion, Yokoyama et al. also found that AC2 and AC6 are more highly expressed in rat DA than in the aorta during the perinatal period [48]. Using AC subtype-targeted siRNAs and AC6-deficient mice suggest that AC6 is responsible for the HA-mediated intimal thickening of the DA, whereas AC2 inhibits AC6-induced HA production [48]. Interestingly, the PGE-EP4 axis was expanded as Iwasaki et al. [39] showed that interleukin-15 (IL-15) inhibits SMC proliferation and HA production in rat DA through attenuation of PGE1-induced HA production in a dose-dependent manner. Taken together, PGE/EP4/cAMP pathway and IL-15 both have essential roles in the regulation of HA production.

#### 4.1.2. Fibronectin (FN)

FN promotes DASMC migration, and its translational regulation depends on a novel target light chain 3 (LC-3). Collecting SMCs from lamb DA at 100 days of gestation, Boudreau et al. found that DASMC migration could be inhibited by administrating peptides that block cell–FN interactions [32]. Diving into FN-dependent SMC migration, other investigators used in vitro cell culture studies to demonstrate that LC-3 overexpression in SMC results in enhanced FN synthesis [49]. These findings suggest that LC-3, a microtubule-associated protein, can act as a RNA-binding protein that enhances FN mRNA translation in DASMCs.

Mason et al. encoded an mRNA decoy to investigate whether binding and sequestering LC-3 may inhibit WT FN mRNA translation [50]. They found that the decoy RNA sequestered LC-3 away from endogenous FN mRNA and inhibited its translation. As LC-3 is sequestered, recruitment of FN mRNA to the polyribosomal translational machinery is inhibited, and FN translation is stalled. Using gene-targeting technology in utero with decoy RNA that transfected DA vessels at the onset of intimal cushion formation, they successfully inhibited intimal cushion formation and retained luminal patency of the DA at the full-term neonatal lambs. The DA transfected with RNA decoy presented with decreased medial ductal thickness, reduced distance to the elastic lamella, and reduced FN in the ECM. This experiment highlights the potentials of gene therapy in targeting matrix components in ductal-dependent congenital heart defects. Furthermore, not only is FN translation implicated in intimal thickening, aberrant posttranslational modifications of FN have been reported to be implicated in the settings of vasculature remodeling of aneurysms and arterial injury [12,51].

### 4.2. Internal Elastic Lamina Disruption

The ultrastructure of the IEL undergoes significant developmental changes towards late gestation with substantial disruptions and impaired elastogenesis [52]. Tada et al. found that a single continuous IEL was still well-defined at 16 weeks’ gestational age; however, at 22–28 weeks, 80% of the DA samples had a duplication and interruption of the IEL in 10–50% of the whole circular structure. And by 29–31 weeks, 60% of the DA was disrupted, followed by 100% of disruption at 32–40 weeks with an interruption of IEL in >50% of the whole circular structure [53]. Hinek et al. have demonstrated that truncated 52 kDa tropoelastin and the reduction of elastin binding protein negatively regulates elastic fiber formation in the DA [35,54]. The inbred of Brown–Norway rats, presenting with hereditary PDA, were associated with an increased vascular fragility mediated by an aortic elastin deficit resulting from decreased elastin synthesis. Bokenkamp et al. reported that PDA is related to the integrity of subendothelial IEL and defective intimal thickening formation [33]. Together with reports on the persistence of the DA with elastic fiber abnormalities in the human PDA [34], these findings suggest that the IEL can limit the passage of SMC from the media to the intima and its disruptions play a role in determining the fate of ductus closure [55]. Moreover, the major morphological difference between the standard mature DA and the persistent DA is the spatially close relationship between EC and the subendothelial IEL, highlighting an altered elastin metabolism in the PDA [34].

#### 4.2.1. LOX

The sparse and truncated properties of DA elastic fibers speculated to contribute to vascular collapse and subsequent closure of the DA after birth [36]. Yokoyama et al. found that EP4 significantly inhibited elastogenesis by decreasing lysyl oxidase (LOX) protein expression, which catalyzes elastin crosslinks in DASMCs but not in Ao SMCs. In EP4-knockout mice, electron microscopic examination showed that the DA relates to an elastic phenotype that was similar to the neighboring Ao. Attempting to establish the hypothesis in the clinical sample, they collected human DA and Ao tissues from seven patients. They found a negative correlation between elastic fiber formation and EP4 expression, as well as between EP4 and LOX expression. Dissecting the molecular pathways behind EP4-mediated elastogenesis inhibition, Yokoyama et al. confirmed that the EP4-cSrc-PLCγ-signaling trail promoted lysosomal degradation of LOX and that together with in vitro experiments, these data suggest that PGE_2_-EP4 signaling inhibits elastogenesis in the DA by degrading LOX protein [7].

#### 4.2.2. t-PA

Tissue-type plasminogen activator (t-PA) expressed in DA EC modulates intimal thickening via activation of matrix metalloproteinase-2 (MMP-2) and subsequent disruption of IEL. Saito et al. reported that an abundantly expressed molecule in EC of the rat DA, t-PA possesses an intrinsic gelatinase activity in converting plasminogen into plasmin that activates MMP-2 [56] and MMP-9 [57], molecules that are recognized in degrading the IEL. Investigating gelatinase activity on a gestational day 21, Saito et al. found an association between the disruption of IEL and marked gelatinase activity, which was inhibited by the MMP inhibitor EDTA. Moreover, gelatin zymography studies demonstrated that pro-MMP-2 was higher in rat DA EC than in Ao EC, with greater MMP-2 activation detected in those of DA. The association of plasminogen was confirmed as supplementation of plasminogen promoted the disruption of IEL and gelatinase activity of MMP. Moreover, t-PA-targeted siRNAs significantly attenuated plasminogen-induced disruption of IEL and gelatinase activity in the 3D vascular models. These data support the concept that the conversion of plasminogen to plasmin promoted IEL disruption via MMP activation. Saito et al. then confirmed their in-vitro findings in human DA from five patients examining t-PA expression using quantitative RT-PCR. In their study, the authors suggest that that t-PA is secreted from DA ECs and that it promotes the plasmin-induced activation of MMP-2 and the subsequent disruption of the IEL, which may contribute to intimal thickening formation in the DA [37].

#### 4.2.3. Carbohydrate and Its Modifications

ECM contains a diverse array of glycan-based structures whose carbohydrate modifications dictate numerous interactions and confer a high degree of selectivity regulating receptor activation [58,59], cell–cell interaction, and cell–matrix interaction [60]. Altered glycosylation of matrix acquires different adhesion and potential of migration properties as the altered conformation of the matrix component may cause a change in available glycan ligands for SMC to bind or activate receptors. Hence reduced cell–matrix interaction may result in reduced migration, growth factor activation and consequent inhibition of intimal thickening. As delineated below, the diverse role of carbohydrate modifications in sequestering growth factors, promoting endothelial–mesenchymal transition, and dampening growth signals brings to light its potential translation to PDA research.

##### Chondroitin Sulfate

Sulfated glycosaminoglycan (GAG) and chondroitin sulfate (CS), were found to modify elastin metabolism in DASMCs. Hinek et al. demonstrated that both N-acetylgalactosamine GAGs, CS and dermatan sulfate (DS) cause the release of 67 kD elastin binding protein (EBP) from the SMC and thus, impairs the assembly of elastin fiber. The authors observed that there is an intrinsic reduction of EBP in the fetal lamb. When neonatal rat Ao SMC was incubated with CS and DS, EBP was reduced through a mechanism of shedding from SMC membranes into the conditioned medium. This process was associated with impaired elastin fiber assembly. With these findings, Hinek et al., propose that an increase of CS or DS impairs assembly of newly synthesized elastin in DA and allows intimal thickening development [35].

##### Biglycan

Biglycan, a small leucine-rich proteoglycan bearing CS/DS/GAG chains and N-linked carbohydrates [61], is normally sequestered in ECM but is proteolytically released during stress or injury. These proteoglycans have been reported to act as a ligand of innate immunity receptors [62], modulate MMP behavior [63], regulate collagen fibril and matrix [64] and interact with various growth factors such as TGF- β, PDGF, and TNF-α [63]. A similar proteoglycan, decorin enhances the PDFG- and IFN-γ- induced proliferative effects in vascular smooth muscle cell (VSMC) [65], and its decoy DS-SILY20 dampens the response through sequestration of the cytokines via matrix [65] and successfully reduced intimal thickening. Early stage of atherosclerosis has been associated with an accumulation of biglycan [26]. The proteoglycan biglycan is regarded as a modifier of remodeling in intimal thickening of vascular injury [66]. Activation of the AT1 receptor, AngII upregulated TGF-β -induced biglycan secretion. Secreted biglycan has been found to interact with numerous matrix components, such as collagen and elastin, and thus become sequestered in the ECM, but liberated during stress through proteolysis. Biglycan has been regarded as the inducer of vascular remodeling in atherosclerosis [66].

##### Perlecan

Several reports have indicated that the basement membrane specific heparan sulfate proteoglycan, perlecan, that has been implicated in modulating VSMC intimal thickening [25,67]. Perlecan was found to induce different cellular phenotypes in various cell types inhibiting proliferation of EC and SMC, an implied to mediate fibrosis and intimal thickening [25]. Sulfate proteoglycan, such as endocan, also has been found for the first time to participate in the process of endothelial–mesenchymal transition [68].

##### Gal-1

N/O-linked glycosylation of Gal-1 protein that binds specifically to b-galactoside sugar suppresses PDGF induced proliferation and migration in a glycan-dependent manner. The migrative phenotype of Gal-1-KO-VSMCs is associated with reduced adherence of VSMC on FN and decreased activation of FAK phosphorylation mediated by increased dephosphorylation and FA disassembly, in a glycan-dependent manner [27]. It is proposed that extracellular Gal-1 protein strengthens VSMC FN interactions and FA stability, and thus limits intimal hyperplasia postvascular injury [27].

### 4.3. ECM-induced Migration of Medial SMC into the SR

Despite evidence accumulating that prior HA deposition promotes SMC migration into the subendothelial layer to form intimal thickening [38], a growing literature has begun unraveling additional mechanisms that ECM components may underlie SMC migration in intimal thickening. This is a process that may not be simply HA-mediated, but mediated by a cascade of signaling repertoire that is tailored to the spatial and functional coordination of its constituency. It has been proposed that in the ductus, increased production of endothelial HA and SMC CS and FN, and the impaired elastin fiber assembly are features critical to SMC migration into the SR and intimal cushion formation.

#### 4.3.1. Hyaluronan Binding Protein

The self-regulation of HA binding protein (HABP) is essential in the successful migration of SMC into the SR. Observing the distinct ECM secretory repertoire of SMCs and ECs of DA compared to that of adjacent Ao and PA, Boudreau et al. aimed to investigate whether these matrix components may induce DASMC or Ao SMC migration seeded on collagen gel. The addition of peptides or antibodies against FN significantly decreased migration in the DA cells but did not affect migration in the Ao. Moreover, the inclusion of HA in the gel further enhanced DASMC migration, while the same enhancement was not observed in Ao SMC migration. The ability of HA in inducing SMC migration was attenuated when treated with antibodies that bind to surface of HABP, a protein synthesized in greater amounts in DASMC compared to that of Ao. These results, together with their previous finding of HA accumulation, suggest that both increased FN and HABP play the roles in enhanced migration of DASMCs and propose mechanisms for increased DASMC migration into the subendothelial matrix [32].

#### 4.3.2. Integrin

ECM receptors on DASMCs must possess the ability to promote cell migration through both interstitial and basement membrane matrices to allow intimal thickening. Integrin receptors expressed on SMCs modulate the migratory properties of the cell, and they enable SMC to adhere to and migrate on ECM components like FN, LN, and collagens I and IV. Clyman et al. found that while SMC adhesion and migration on these substrata are entirely dependent on the presence of functioning f31 integrins, cell migration depends on both f31 and f33 integrins [69]. Furthermore, ligand affinity chromatography and immunoprecipitation techniques identified a distinct pattern of integrins binding to each matrix component: FN (alpha 5 beta 1, alpha 3 beta 1, alpha V beta 1), LN (alpha 1 beta 1, alpha 7 beta 1), vitronectin (VN) (alpha V beta 1), I (alpha 1 beta 1, alpha 2 beta 1), and IV (alpha 1 beta 1). In contrast, the beta 3 integrin, alpha V beta 3, bound to all the substrates tested: FN, LN, VN, I, and IV. The results indicate that beta 1 and beta 3 integrins may play different roles in attachment and migration as SMC moves through the vascular ECM to produce obliteration of the DA lumen [69].

#### 4.3.3. TGF- β

Exposure of DASMC to TGF-β increased the formation of focal adhesion (FA) plaques that are the integrin-containing, multiprotein structures that act as tight linkers between the cell’s intracellular actin bundles and matrix substrate. Tannenbaum et al. found that TGF-β appears to increase the ability of integrin receptors to associate ECM tighter with the cytoskeleton, increasing the cell’s adherence to the ECM and thus, limiting their mobility [40]. Levet et al. dissected into two members of the TGF-β family: bone morphogenetic proteins BMP9 and BMP10, and found that engineered Bmp9 knockout mice increased the event of an imperfect DA closure [42]. Administrating neutralizing anti-BMP9 antibodies to postnatal Bmp9 knockout mice exacerbated the closure defects and promoted DA reopening. Transmission electron microscopy images revealed that these defects were associated with dedifferentiation of endothelial to mesenchymal cells and gross reduction of matrix deposition at the lumen. They further confirmed that human genomic data could define a critical region in chromosome 2 encoding genes including BMP10 that correlated with the presence of PDA [42]. Together, these data establish roles for BMP9 and BMP10 in DA closure. Additionally, literature has reported that TGF-β enhances matrix production and remodeling when added to cultured SMCs [40,70].

#### 4.3.4. Peroxidasin

ROS has long been established to induce vascular remodeling and VSMC migration. Considering the abrupt exposure to the rise of oxygen during the fetal–neonatal transition, activation of oxidation pathways plays a significant role in intimal thickening. Peroxidasin has been identified to be elevated in vascular walls that undergo remodeling, and knockdown of peroxidasin reduces Ang II-induced VSMC proliferation. As AngII belongs to the PGE2-EP4 axis that has been previously established in PDA, Shi et al. found that VSMC deficient in EP4 increased AngII-elicited vasoconstriction and pathological vascular remodeling [24]. Yang et al. further elaborated the role of peroxidasin in cardiac hypertrophy and found that peroxidasin mediated AngII-induced cardiac hypertrophy via the Nox2/VPO1/HOCl/ERK1/2 redox signaling pathway where Ang II increased the hypertrophy-related gene (BNP/ANF) expression [71]. Although the peroxidasin concept has not been established in PDA, inhibition of its downstream AngII with BNP has prevented proliferation and migration of DASMCs through reducing mitochondrial ROS production [72]. The overlapping pathways of peroxidasin in the DASMC phenotype suggest that peroxidasin can be a promising therapeutic target for clinical management of DA patency.

Moreover, peroxidasin has been previously discovered in lower organisms to stabilize or seal ECM by collagen IV crosslinking [73,74,75], and such a genetic defect is also implied in the human setting [76,77], suggesting an exciting target to study in the role of matrix defects in the pathogenesis of PDA [78,79].

#### 4.3.5. Focal Adhesion

FA not only bridges cell–matrix communication in vascular development but also plays a role in cellular repair. ECM proteins associated with focal adhesions and their integrin receptors have been documented at the levels of homing, tissue organization, and differentiation. Targeted gene deletion studies have demonstrated the critical roles that FN and a5b1 integrin serve in stabilization and branching morphogenesis during vascular development in the murine embryo [80]. FA mediates bidirectional crosstalk between ECM and the cytoskeleton, where biochemical and mechanical cues are exchanged. This ECM signaling directs rearrangements in cytoskeletal organization and nuclear gene expression in response to changing conditions in the circulation [81].

### 4.4. Obliteration of the Lumen

The changes in the ECM that induce neointimal formation are also observed in pathological conditions in pulmonary and coronary arteries. The plasticity of the pulmonary circulation in the perinatal period also involves matrix regulation. Processes that prevent the regular decrease in pulmonary vascular resistance have been proposed to impair matrix regulation and cause aberrant developmental and structural changes in the PA. These changes in development include abnormal SMC differentiation, hypertrophy, and proliferation, which exacerbate high pulmonary artery pressure. Degeneration of SMC is usually observed ≥5 days post-delivery [82] following DA functional closure. Accumulation of SMC undergoing cystic necrosis in the closing DA is proposed to be caused by ischemia of the constricted vessel wall. Literature has also shown that apoptotic cells, rather than necrotic cells, are frequently seen in areas of cystic necrosis. Goldberg et al. demonstrated that the combination of hypoxia and hypoglycemia increased apoptosis in lamb DA tissues [83]. Furthermore, isolated instances of apoptosis were also reported in the area of intimal thickening after birth [84].

During the process of obliteration of the lumen, the intima and media gradually become richer in elastic and collagen fibers [52,81]. Collagen fibrils that are frequently absent in the media of the midgestational DA are abundantly expressed in postnatal DA tissues with intimal thickening from a one-day-old infant. The DA lumen is progressively occluded by luminal fibrosis for months following anatomical closure. After the complete occlusion of the DA lumen, the structure becomes known as ligamentum arteriosus, where central zones are collagenized and focally calcified [85]. In summary, we illustrate a conceptual review in Figure 3 for what we have discussed in this review.

## 5. Conclusion and Future Prospects

Despite accumulating associations of ECM with PDA pathophysiology, only a few studies have directly investigated ECM interactions or production. Thus, it is necessary to design more studies that directly intervene with ECM to distinguish comprehensively whether matrix differences arise merely as a result of the disease or act as the driver component underlying the disease to confirm whether ECM abnormalities are directly responsible for DA patency. Moreover, prematurity has long been regarded as a high-risk factor for PDA incidences. Nevertheless, no association has been made between prematurity and possible subsequent ECM consequences. How the physiological factors may be related to prematurity intertwine with abnormalities of DA ECM remains to be elucidated.

ECM proteins are especially intriguing in that they play a multifaceted role not only in their signaling repertoire but also the spatiotemporal design, turnover, and constituency of the scaffold configuration that are tailored for the specialization of the tissue. Moreover, spatial and functional coordination in response to inputs from the bloodstream and surrounding parenchyma are all conditions that may be overlooked in most of the studies that are performed in vitro. Nevertheless, the molecular diversity and plasticity of ECM present a rich array of potential therapeutic targets for the management of vascular dysfunction and disease.

## Figures and Tables

**Figure 1 ijms-21-04761-f001:**
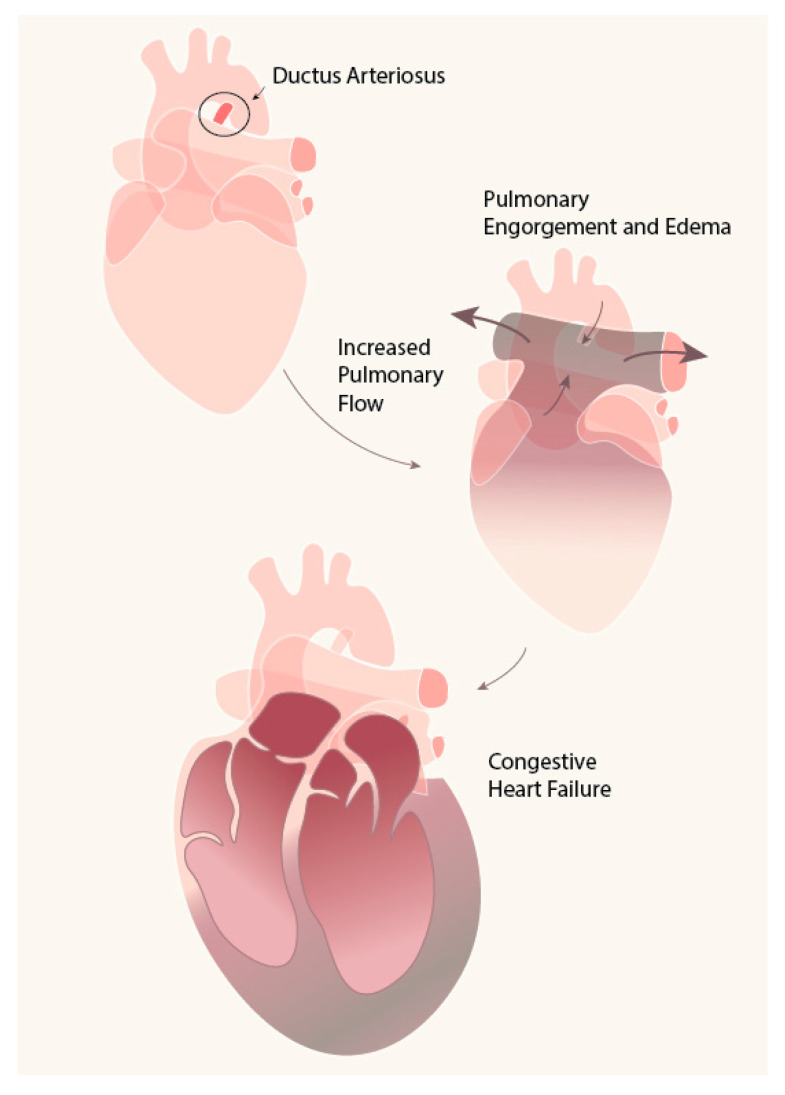
Pathophysiology of hemodynamic burden in patent ductus arteriosus (PDA). Blood from the high-pressure aorta shunts to the low-pressure pulmonary artery, causing pulmonary hyperemia.

**Figure 2 ijms-21-04761-f002:**
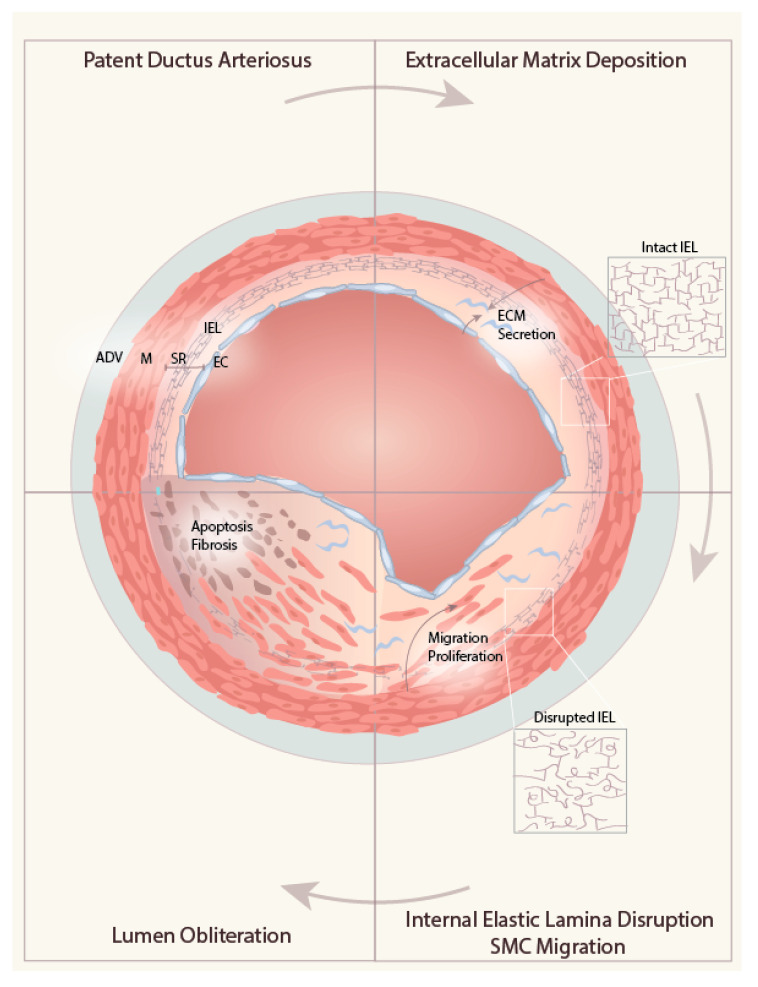
Sequential steps required for intimal thickening of ductus arteriosus closure. Complete closure is sequentially mediated in four phases: the deposition of the extracellular matrix in the subendothelial region, the disruption of the internal elastic lamina (IEL), followed by the migration into the subendothelial space of undifferentiated medial smooth muscle cell (SMC) for the formation of intima thickening. Finalizing the closure by luminal obliteration, SMCs undergo apoptosis and fibrosis to form ligamentum arteriosus. ADV: adventitia, M: media, SR: subendothelial region, EC: endothelial cell, IEL: internal elastic lamina, ECM: extracellular matrix.

**Figure 3 ijms-21-04761-f003:**
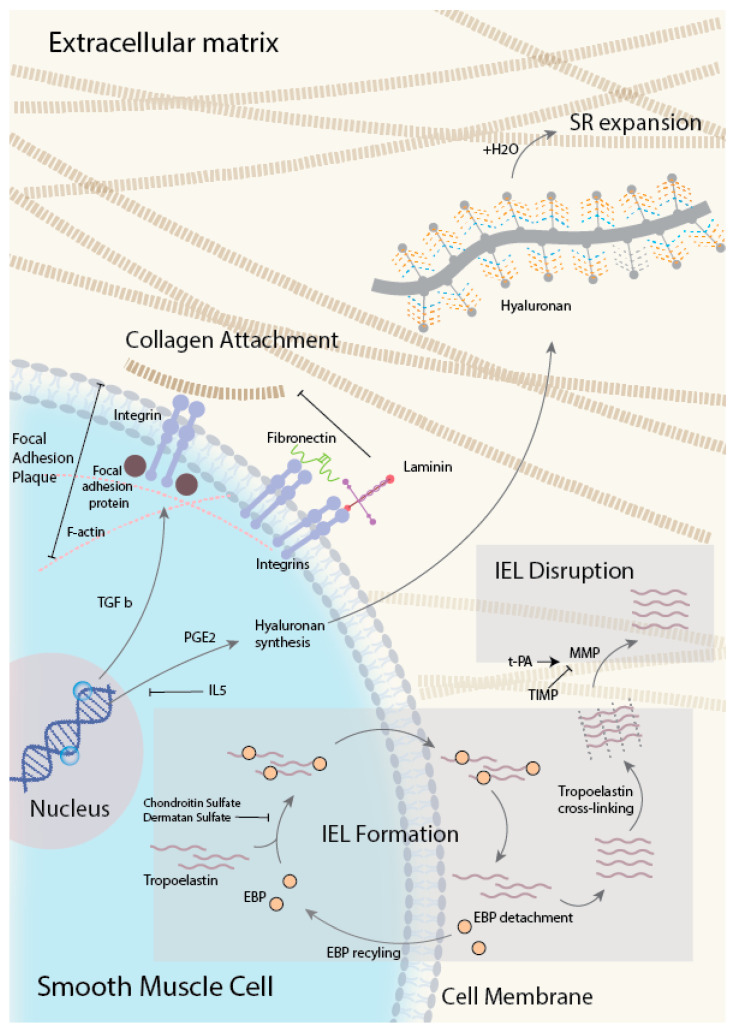
A conceptual overview of the role of ECM in DA. Integrin acts as an adhesive to the ECM matrix and promotes the migration of SMC into ECM through adhesion. Hyaluronan promotes H2O influx into the subendothelial region (SR) of which causes the region to loosen and expand and promote SMC migration. Degradation of tropoelastin crosslink IEL reduces the barrier integrity and allows SMC to migrate through and into the SR. MMP: matrix metalloproteinase, TIMP: tissue inhibitors of metalloproteinase, t-PA: tissue-type plasminogen activator, EBP: Elastin binding protein.

**Table 1 ijms-21-04761-t001:** Role of extracellular matrix (ECM) in systemic vascular intimal thickening.

Matrix Element	Effect on SMC	Mechanism	Reference
Glycoprotein			
Fibronectin	Proliferation	Cyclin activation	[12]
Vitronectin	Migration	plasminogen activator inhibitor-1	[13]
Proteoglycan			
Heparan sulfate	Anti-proliferation	Growth factor interaction	[14,15,16]

**Table 2 ijms-21-04761-t002:** Role of ECM-related mediators in systemic vascular intimal thickening.

ECM Mediator	Effect on SMC	Mechanism	Reference
Matrix binding receptors			
Focal Adhesion	Migration	Increased migration through ECM adhesion	[17]
Vitronectin receptor	Migration	Increased SMC accumulation	[18]
Matrix degrading enzymes			
TIMP	Anti-migration	MMP inhibition	[19]
MMP	Migration	Matrix degradation	[20]
PDGF	Migration	MMP expression	[21]
t-PA	Migration	Matrix degradation	[22]
LOX	Anti-migration	Matrix crosslink	[23]
Peroxidasin	Anti-migration	Matrix crosslink	[24]
Carbohydrate Modification			
Perlecan	Anti-proliferation	unknown	[25].
Biglycan	Pro-inflammation	unknown	[26]
Gal-1	Anti-proliferation	suppresses PDGF induced response	[27]
Matrix mediating cytokine			
TGFβ_1_	Matrix production Proliferation	Fibronectin and collagen production, DNA synthesis	[28]
Thrombin	Migration	MMP inhibition	[29]

**Table 3 ijms-21-04761-t003:** Roles of matrix elements in ductus arteriosus (DA) remodeling.

Matrix Element	Effect	Mechanism	Reference
Glycoprotein			
Fibronectin	Anti-adhesive	Cytoskeletal reorganization	[31,32]
Laminin	Anti-adhesive	Inhibit SMC binding to collagen	[31]
Elastin	Elastin metabolism	Reduced IEL fragmentation enforces barrier integrity against migration	[33,34]
Proteoglycan			
Chondroitin sulfate	Elastin assembly	Release elastin binding protein reduces decreases elastin fiber assembly	[35]
Dermatan sulfate	Elastin metabolism	Release elastin binding protein reduces decreases elastin fiber assembly	[35]
Glycosaminoglycan			
Hyaluronan	Migration	The influx of water loosens and expands the subendothelial region, SMC binds to hyaluronan through hyaluronan binding protein	[32,36]

**Table 4 ijms-21-04761-t004:** Overview of different matrix mediators and how they may influence DA closure.

ECM Mediator	Effect	Mechanism	Reference
Matrix binding receptor			
Integrin	Adhesion Inhibit migration	Increases SMC adhesion to LN	[31]
Matrix degrading enzymes			
t-PA	Elastin metabolism	Increases MMP-2 and MMP-9 expression that promotes elastic laminae degradation	[37]
LOX	Elastin formation	Catalyze elastin crosslink	[23]
Matrix production cytokines			
PGE2	Hyaluronan deposition	Induces hyaluronan synthase type 2 mRNA	[38]
IL-5	Inhibit proliferation, matrix production	Decreased SMC proliferation and hyaluronan production	[39]
TGFβ_1_	Adhesion	Increased focal adhesion plaque formation and integrin receptors expression	[40,41]
BMP9,10	Differentiation, matrix production	Bmp9 knockout in mice led to imperfect closure of the DA. Promotes intimal cell differentiation, ECM deposition	[42]
Others			
Tropo-elastin	Elastin formation	Decreases elastin binding protein expression	[35]

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
