# Peer review of "Role of Extracellular Matrix in Pathophysiology of Patent Ductus Arteriosus: Emphasis on Vascular Remodeling"

_ijms, 2020, doi:10.3390/ijms21134761_

Round 1
Reviewer 1 Report
The review presented by the Authors is comprehensive and easy to read even by someone like me who, though being in the extracellular matrix field, has no knowledge of cardiology nor of the specific focus of the work (Patent Ductus Arteriosus), which is a sure plus of the whole work.
The only element I found worrisome is that the literature is somewhat outdated, with practically all the reviews being 10 to 20 years old. Again, I am not expert in this specific field so I´m not able to judge whether these somewhat old references still represent the leading edge of the field. I would, however, doubt it given the progress of ECM/matrisome filed in many other fields.
Thus, possibly, the Authors should undergo another round of review of the literature to make absolutely certain that their piece stays on top of the current knowledge before resubmitting.
A very minor point, also, is that the term Patent Ductus Arteriosus should be introduced in extenso in the abstract before the first abbreviation.
Reviewer 2 Report
The review by Lin, Yeh and Hsu is a very approachable summary of Ductus Arteriosis (DA) from the perspective of ECM function and pathology, and follows up on their 2018 review that focuses more on signaling pathways. The paper should provide a useful introduction to those interested in, or new to DA. It is noteworthy that only 3 references date from 2015 or later- including the previous review by the authors. There is little introduced here that is new. The recent literature on DA is pretty quiet on ECM aspects – perhaps this is a call by authors to revisit this phenomenon? If so, the authors should be more explicit, and identify issues worthy of study with the many genetic tools that have emerged in the last 20 years. There is great progress on ECM remodeling in other tissues and models that could be brought back to DA -such as carbohydrate modification, or ECM crosslinkers like peroxidasin, glycans and perlecans that could be extended to this disease model. It would be good for the authors to acknowledge or explore this.
There are many minor issues with language and organisation of the manuscript:
-Acronyms are inconsistently defined- some not at all, and many not at first use
-Table 2 is missing lox, peroxidasin, and carbohydrate modifications
-paragraph starting on line 190 is muddled and confusing
-These lines have grammar issues- including recurrent trouble in matching plurals:
78, 132, 134, 138, 145, 166, 187, 194, 201, 215, 335,
Round 2
Reviewer 1 Report
All my concerns have been addressed. I thus suggest acceptance of this manuscript.